# Unfavorable Behaviors in Children Run in Packs! Dietary and Non-Dietary Modulators of Attentional Capacity

**DOI:** 10.3390/nu14245264

**Published:** 2022-12-09

**Authors:** Alina Drozdowska, Michael Falkenstein, Thomas Lücke, Mathilde Kersting, Gernot Jendrusch, Petra Platen, Kathrin Sinningen

**Affiliations:** 1Research Department of Child Nutrition, University Hospital of Pediatrics and Adolescent Medicine, St. Josef-Hospital, Ruhr University Bochum, 44791 Bochum, Germany; 2ALA Institute, 44805 Bochum, Germany; 3Department of Sports Medicine and Sports Nutrition, Ruhr University Bochum, 44801 Bochum, Germany

**Keywords:** attention, breakfast, sleep, physical activity, behavior

## Abstract

Children’s cognitive performance can be influenced by behaviors such as eating breakfast in the morning. The aim of this analysis was to investigate the mediating effects of breakfast behavior and other lifestyle habits on eye-hand coordination and attention. In a secondary analysis of the CogniDROP study, children from the 5th and 6th grade of a comprehensive school in Germany (*n*  =  223) performed a simple computerized Visual Attention Task and answered a questionnaire about behavioral patterns, i.e., skipping breakfast on a school day, frequency of physical activity (PA) outside school, and nighttime sleep. An association matrix was constructed to show the relationship between the variables. Almost 11% of children left home in the morning without breakfast, more than 9.5% of children reported poor sleep quality, 24.9% slept less than the recommended 9 h, and girls were insufficiently physically active. Sleep duration, bedtime, and PA correlated with skipping breakfast. Better sleep quality was positively related to reaction time in the Visual Attention Task. Overall, the data suggest that unfavorable behaviors in children tend to run in packs, just as skipping breakfast in the morning seems to be associated with other unfavorable habits, which impairs children’s eye-hand coordination and attention.

## 1. Introduction

There is scientific evidence of the influence of children’s behavioral patterns on their health and cognitive functions [1,2,3,4,5]. In the short term, breakfast consumption has a positive effect on cognition, including attention, memory, and executive functions compared to fasting [2]. However, breakfast is the most skipped meal among children, especially girls [2,6]. This increased the focus on skipping breakfast in children, especially in relation to screen use [7] and overweight [8].

Another factor whose effects on cognition have been studied is sleep. Adequate sleep duration is associated with better cognitive functioning and physical health among children [1,3]. Accordingly, it is recommended for school children up to the age of 12 to regularly sleep 9–12 h per night to promote their mental health [9]. Lack of sleep is associated with obesity and sedentary activities such as media screening [1,5], possibly leading to daytime sleepiness [10] and cognitive impairment [3]. In general, recreational screen use among children and adolescents has increased during COVID-19 compared to pre-pandemic levels, while physical activity (PA) has decreased. Before the pandemic, the prevalence of physically inactive students was around 20%, while leisure time screen use was around 2.5 h per day with age- and country-specific differences [11]. Recently, other authors have shown that prolonged use of digital devices in children between 10 and 12 years of age is related to shorter sleep and less PA. Overall, each of the three habits correlated with attention and working memory [12]. Unfavorable lifestyle habits are the main cause of various civilization diseases, including obesity and mental disorders [8,9]. Understanding their interaction could facilitate effective prevention at a young age.

The influence of individual unfavorable habits on children’s cognitive performance is well known. What about behavioral patterns? Do multiple unfavorable habits add up to even worse cognitive outcomes? Given the high prevalence of unfavorable habits among school-aged children, there is growing interest in further prevention efforts against this development. Therefore, more attention should be paid to the relationships between these behaviors. Most studies have not examined the association between breakfast and sleep [2,6,7,8,13,14], especially considering sleep quality, attention capacity, and PA as additional variables [5,15,16,17]. Therefore, the aim of this study was to investigate the relationship between lifestyle characteristics such as breakfast intake, sleep, and PA with performance on an attention task. The question of the study was whether certain behaviors accumulate in children and correlate with cognition. It is hypothesized that skipping breakfast is related to other unfavorable behaviors in children. For this specific objective, data from the CogniDROP study (Cognition, Drinking Observation and Physical Activity) were used as a secondary analysis [18,19].

## 2. Methods

### 2.1. Study Design and Recruitment

The CogniDROP study was approved by the ethics committee (the Ethics Committee of Ruhr University Bochum, Bochum, Germany; EKS V 22/2018). Participant recruitment and data collection took place between October 2018 and January 2019 during regular school days. Fourteen 5th and 6th grade classes at a comprehensive school in Gelsenkirchen, Germany took part in the study. In this secondary analysis, cross-sectional data are presented and analyzed, neglecting randomization within the CogniDROP intervention study (described elsewhere [18]).

### 2.2. Study Schedule and Participants

Of the 279 eligible children with parental consent, 223 children were present on the day of data collection and were included in the analysis. Children with learning difficulties participated in the study, but data from those children were not included in the analysis. In addition, personal data, i.e., sex, age, and Body Mass Index (Seca 862 digital scale, Seca Corporation, Hamburg, Germany) were recorded. Regular school lessons began at 8:00 a.m. and breakfast break at 9:15 a.m. Drinking water and food intake were allowed. On the study day, a computerized Visual Attention Task was performed at 12:15 p.m., after regular classes, and a questionnaire on habits (sleep, breakfast, non-school PA) was answered. The term non-school PA included sports activities outside of school, with or without supervision by a trained teacher. No physical education in school took place in the last 24 h before the cognitive task to avoid possible bias. For this period, participating school children were given an accelerometer to wear on their wrist for 24 h to measure their PA (ActiGraph GT3X, Pensacola, FL, USA). Times of sedentary behavior in percent (sedentary within 24 h) and level of PA based on the number of steps (step counts within 24 h) were evaluated.

### 2.3. Cognitive Assessment

In a quiet classroom, a computer-based Visual Attention Task was performed within each class. This simple task aimed to determine how familiar the children were with the computer mouse and how attentively they followed the task. Furthermore, this task was designed as a preliminary exercise in the CogniDROP study to allow children to train eye-hand coordination for later complex cognitive tasks. This approach, used in studies with preschoolers with and without developmental disabilities, can help children become familiar with the computer mouse [20,21]. In this task, 26 white squares were displayed on a black screen that turned green consecutively and dimmed after a correct answer was given (Figure 1A). The children should click on the green squares as quickly and accurately as possible within 3 min. The response time was evaluated at the beginning (First RT), halfway through the task (Half-time of RT), and at the end of the task (Total RT). The computerized task was finalized with the questions: What time did you go to bed last night?; What time did you get up this morning?; Did you sleep well? (sleep quality); Did you have breakfast at home this morning?; and How often do you do sports outside of school? (non-school PA). An example of questions and answers is presented in Figure 1B.

### 2.4. Statistical Analyses

The detailed description and calculation of the sample size is described elsewhere [18]. At least 51 participants were required for each group difference analysis using the statistical software package IBM*SPSS* Statistics for Windows, version 25.0 (IBM Corp., Armonk, NY, USA). The level of significance was set at *p* ≤ 0.05. Distributions were tested for normality with the Shapiro–Wilk test. Characteristics of participants were reported as frequencies and percentages for categorical data. Response time on the Attention Visual Task and breakfast characteristics were the primary outcomes of the study. Breakfast behavior was stratified as a) skipping breakfast at all, b) did not eat or drink, and c) having breakfast/eating and drinking. These three categories were used for group comparisons with the Kruskal–Wallis test for non-normally distributed data. For pairwise comparisons between breakfast groups, Dunn’s test with a Bonferroni correction was performed. PA was recorded either as an objective measure (continuous variable: number of steps and sedentary activity within 24 h as a percentage via ActiGraph) or subjectively by self-report (categorical variable: non-school PA as frequencies). Sleep quality was categorized into poor and good sleep quality. Differences between boys and girls were tested with an independent samples *t*-test for normally distributed data or the Mann Whitney U Test for non-normally distributed data. Linear regression model was conducted to determine the relevant predictors (breakfast, BMI, sleep, PA) of response time in the Visual Attention Task. Supplementary, Spearman’s rank correlation coefficient was used to graphically visualize the relations between the studied variables (Figure 2).

There was a positive correlation between wake up-time and sedentary behavior (*p* = 0.006, *r* = 0.188), suggesting that children who were more sedentary woke up later on the test day. Other characteristics of sleep and PA did not correlate (*p* > 0.05).

## 3. Results

The participants’ characteristics are presented in Table 1. On average, 10.9% (*n* = 24) of the children did not eat breakfast at all before going to school and 6.3% (*n* = 14) did not do sports outside school. The children slept more than 9 h on average and more than 9.5% (*n* = 21) of the children reported poor sleep quality. Fifty-five of 221 (24.9%) children slept less than the recommended 9 h for this age group [9].

### 3.1. Differences between Boys and Girls

Sex differences were found only for PA as well as for response time on the Visual Attention Task. Girls did fewer sports outside of school than boys, and over 12% of girls did no sports at all.

For the more than 15,000 steps within 24 h, the boys needed significantly less time (sedentary behavior was over 46%) than the girls (sedentary behavior was over 43%). The boys also showed better response time when using the computer mouse at the beginning (First RT) and later, resulting in faster performance on the overall Visual Attention Task than the girls (Table 1). Considering separately by sex, BMI was inversely correlated with PA only in boys (*p* = 0.030, *r* = −0.194), but not in girls (*p* = 0.183). In general, BMI correlated only with non-school PA, i.e., BMI decreased when PA increased (*p* = 0.003, *r* = −0.210).

### 3.2. Breakfast Habits

Significant differences were found for the three breakfast categories on the variables of sleep and PA (Table 2). Children who slept longer and went to bed earlier were more likely to have breakfast at home. There was also a significant association between skipping breakfast and lower activity (fewer steps and less frequent non-school PA). After applying the Bonferroni correction, the significant differences in sleep duration remained between the variables “had breakfast” and “did not eat or drink” (*p* = 0.004). Furthermore, bedtimes were significantly different between “had breakfast” and “did not eat or drink” (*p* = 0.006) and between “had breakfast” and “skipping breakfast” (*p* = 0.030). Significant differences in non-school PA were found between “had breakfast” and “did not eat or drink” (*p* = 0.019) and between “had breakfast” and “skipped breakfast” (*p* = 0.025). Significant differences in the number of steps were found between “had breakfast” and “skipping breakfast” (*p* = 0.013) and between “did not drink or eat” and “skipping breakfast” (*p* = 0.018). Skipping breakfast did not correlate with poor sleep quality (*p* = 0.376), nor with BMI (Table 2). Sleep quality also did not correlate with all other variables (*p* > 0.05), except for response time in the computer task (Figure 2).

### 3.3. Visual Attention Task

The range of response time at the beginning (First RT) was between 2.6 s and 21.0 s, and at the end of the task (Total RT) was between 25.0 s and 78.0 s. Children who were faster at the beginning also finished the task faster (*p* < 0.001, *r* = 0.494).

For the regression model constructed with response time in the cognitive task as the dependent variable and the predictors (breakfast, BMI, sleep, PA), the model did not reach statistical significance (F(5.176) = 1.533, *p* = 0.182). Therefore, further analyses were based on bivariate correlations. All results are shown in the graph (Figure 2). In general, the graphical illustration shows how a 24-h behavioral pattern of children significantly influences the cognitive task (black arrows).

There was no significant correlation between BMI, wake up-time, as well as sleep duration with response time in the visual task (*p* > 0.05). However, poor sleep quality was related to slower total visual response time (for Total RT, *p* = 0.041, *r* = 0.138) and late bedtime correlated with faster response time (for First RT, *p* = 0.017, *r* = −0.161, for Half-time of RT, *p* = 0.018, *r* = −0.158, for Total RT, *p* = 0.031, *r* = −0.145).

Children with longer sedentary behavior in the previous 24 h responded faster at the beginning of the computer task (for First RT, *p* = 0.028, *r* = −0.149), regardless of the number of steps. However, this did not affect the overall performance on the test (*p* > 0.05). Although the number of steps correlated with sedentary behavior (*p* < 0.001, *r* = −0.522), there was no significant correlation between the number of steps and response time (*p* > 0.05). Skipping breakfast at home did not correlate with cognitive response in the task (Table 2).

## 4. Discussion

The purpose of this study was to evaluate the lifestyle behaviors, specifically breakfast intake, of school children in relation to the Visual Attention Task. The results showed that certain behavioral patterns accumulate in children and appear to influence cognition. However, skipping breakfast in the morning did not correlate with response time in the attention task at noon. In line with other studies, the results indicate associations between skipping breakfast and PA [6,22], as well as between skipping breakfast and sleep [15]. Thus, it can be assumed that skipping breakfast is related to an unfavorable behavior pattern the night before [11,16,17].

Consistent with other countries, this study showed that over 10% of children did not eat or drink at all before school [6,8,15]. However, if breakfast is not eaten at home, it can be done during breakfast at school, which in turn provides necessary nutrients and eliminates possible attention deficits [2,22]. This may have reduced the association between breakfast behavior and reaction time in the computer task in this study. However, unhealthy snacks at school can have a negative effect on the body [15,22]. Because the children’s morning eating patterns were not documented at school, no conclusions about nutrient intake could be drawn.

While other studies have reported sex differences and an influence of breakfast on BMI, no differences between girls with boys and no associations of skipping breakfast and BMI were documented here [6,8,16,17]. This could be since BMI is influenced in the long-term and not the result of skipping breakfast once or of other occasional behaviors. In addition, sex differences may be more pronounced after puberty than in children up to 12 years of age [16,17,23].

Several factors may influence whether breakfast is eaten [24]. In this study, a correlation between sleeping and breakfast was shown. While sleep quality and wake up-time did not correlate with breakfast habits, children with shorter sleep durations and later bedtimes were more likely to skip breakfast. Based on studies with children, it can be assumed that a later bedtime is associated with later meals or snacks [5,15]. It is possible that this reduces morning hunger [24]. In addition, bedtime was not related to sleep quality, but poor sleep quality was associated with slow reaction time in the attention task. Others pointed out that poor sleep quality affects children’s mental health and cognitive functions [14]. However, it should be emphasized that the assessment of sleep quality in relation to cognitive function and sedentary behavior is lacking in studies of school-aged children [3,16,25]. For future studies, it may be important to consider not only children’s bedtime and sleep duration. In this analysis, the cause of poor sleep quality could not be explained by documented parameters. Potentially, a larger sample size (at least 51 participants for group comparisons) could affect the statistical power and thus the results.

Furthermore, the results support the notion that a lack of PA is associated with skipping breakfast and a higher BMI. Children who were regularly more active outside of school and had taken more steps in the previous 24 h were also more likely to eat breakfast at home. Such associations have been observed elsewhere as well [6,8,22]. For this reason, exercise can help not only with weight management, but also with changing breakfast habits. There was one unexpected finding, however. After sex-specific analysis, the association between PA and BMI vanished in girls. In addition, no BMI differences were found between girls and boys, although results indicated a lower frequency of non-school PA in girls compared with boys. In contrast, boys spent more time sedentary than girls in the past 24 h. Thus, the results underline the sex differences in PA observed elsewhere [8,16], but do not explain the lack of association between BMI and PA in girls. It may seem that girls need more support for PA, but not to reduce their weight. However, boys seem to benefit more from PA to maintain a healthy weight. It was also interesting to note that despite the lack of differences in step counts between the sexes over the past 24 h, boys were sedentary longer during this period. This could indicate a more vigorous and faster activity of the boys in counting steps. Considering all these results, it highlights the importance of studying girls and boys separately. Thus, sex differences in behavioral analysis could play a greater role in future research.

Boys spend more time sitting at the computer than girls [16]. Therefore, it is possible that girls’ slow reaction time on the computer task in this study may result from poorer eye-hand coordination and less experience with computers. Since no other leisure activities were asked in this study, it is not possible to fully explain the reasons for the boys’ faster reaction time in the computer task. However, other authors suggest that boys respond faster than girls in simple attention tasks [26]. In difficult tasks, the advantages of proper eye-hand coordination may no longer be relevant.

Another finding was that some unfavorable behaviors were associated with improved eye-hand coordination, expressed in the first RT and total reaction time in the computer task. For example, children with longer sedentary behavior and children who went to bed late showed better eye-hand coordination at the start of the computer task; even a late bedtime correlated with faster reaction time on the overall task. Based on previous findings that longer screen time in school-aged children correlates with faster reaction times in computer-based visual search [12], it is possible that in this study the children with better eye-hand coordination had more computer experience. Considering the large response span between 2.6 s and 21 s to start the computer task, it is possible that the children had different experiences with the computer mouse. Therefore, it cannot be explained whether the better attention during the computer task resulted from better eye-hand coordination or better cognitive abilities. Thus, performance on the computer-based cognitive tasks could be affected by irregular computer use, leading to misinterpretation of the results regarding attentional performance.

### Limitations

There were several limitations of the present study. The small number of subjects in the specific groups, i.e., skipping breakfast and poor sleep quality, may result in lower power to detect significant correlations. Recording food consumption (type, quantity, and time) during school morning and questioning the reasons for skipping breakfast could help to better describe breakfast habits. Data on children’s computer experience and screen time would be helpful in explaining the sex differences in the attention task. It should also be mentioned that self-reported children’s behaviors may contain some biases.

## 5. Conclusions

Breakfast consumption before school was associated with a higher number of steps taken in a 24-h period, frequency of physical activity outside of school, sleep duration and bedtime in school-aged children. The children who ate breakfast went to bed earlier. Sleep quality, but not sleep duration, could be important for the performance on the Visual Attention Task. Sedentary behavior had different associations that were positive for the computer task but negative for wake up-time. Several sex differences were found that require further investigation. The acquisition of computer experience should be considered when planning the study design with computer-based cognitive tasks. Taken together, one unfavorable behavior rarely comes alone; thus, holistic approaches are needed to promote children’s cognitive performance and well-being.

## Figures and Tables

**Figure 1 nutrients-14-05264-f001:**
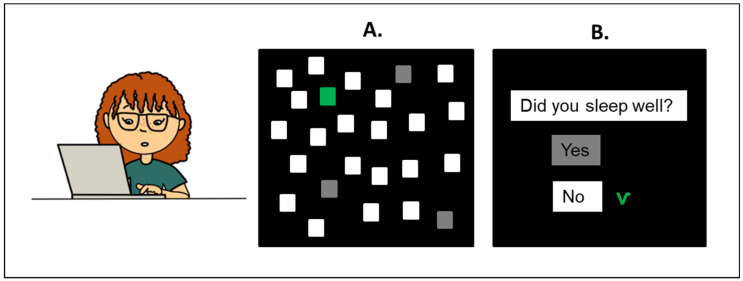
**Computer tasks:** (**A.**) Visual Attention Task. 3-min task to test the response time. The green squares had to be clicked consecutively with the mouse cursor. (**B**.) Behavior questionnaire (an example of questions and answers).

**Figure 2 nutrients-14-05264-f002:**
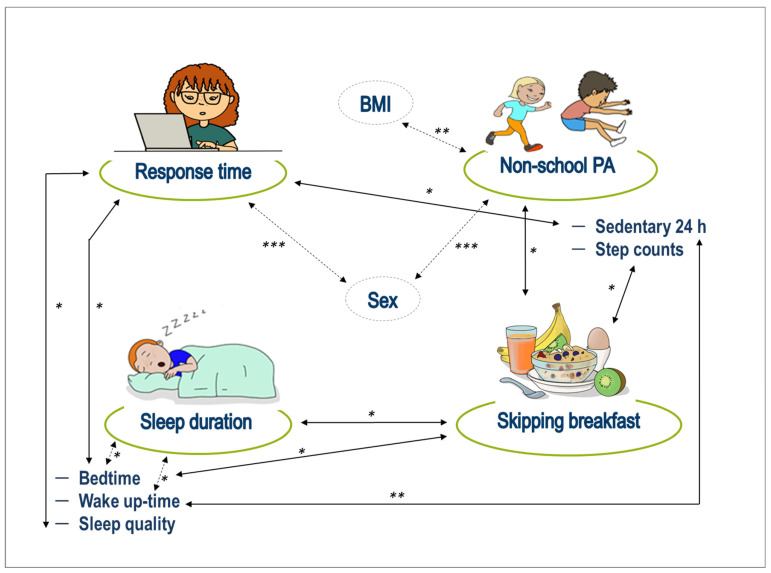
Relations between the studied variables based on the Kruskal–Wallis test and Spearman’s rank correlation coefficient. Skipping breakfast and non-school PA were entered as categorical variables; response time in the Visual Attention Task, step counts, sedentary, sleep duration, bedtime, wake up-time, and BMI were entered as continuous variables. Sex and sleep quality were entered as dichotomous variables. The arrows show significant correlations between behavioral characteristics. BMI = Body Mass Index, PA = physical activity. * *p* ≤ 0.05, ** *p* < 0.01, *** *p* < 0.001.

**Table 1 nutrients-14-05264-t001:** Participants’ characteristics (*n* = 223).

	Boys	Girls	*p* Value
***n* (%)**	142 (63.7)	81 (36.3)	
**Grade 5**	67	34	
**Grade 6**	75	47	
**BMI**	20.1 ± 4.8	20.4 ± 4.3	0.283
**Sleep**			
Duration, hours	9:15 ± 1:14	9:10 ± 1:26	0.691
Poor sleep quality, *n* (%)	13 (9.2)	8 (10.0)	0.817
Bedtime	20:47	20:55	0.596
Wake up-time	06:17	06:12	0.203
**Breakfast**			
Had a full breakfast, *n* (%)	92 (65.2)	43 (53.8)	0.087
Skipping breakfast at all, *n* (%)	13 (9.2)	11 (13.8)	
Did not eat or drink, *n* (%)	36 (25.5)	26 (32.5)	
**Physical Activity**			
Non-school PA frequency	4 times/week	2 times/week	<0.001
No PA at all, *n* (%)	4 (2.8)	10 (12.5)	<0.001
Sedentary within 24 h, %	46.1	43.4	0.026
Step counts within 24 h	15,826	15,618	0.947
**Visual Attention Task**			
First RT (s)	4.1 ± 1.7	4.5 ± 1.2	<0.001
Half-time of RT (s)	20.8 ± 4.0	23.3 ± 5.0	<0.001
Total RT (s)	38.1 ± 7.2	43.0 ± 9.3	<0.001

Descriptive analysis of the dichotomous variable (sex) by Mann–Whitney U test and *t* test; data are presented as median or average values ± standard deviation; *p* ≤ 0.05. *n* = number; PA = physical activity; RT = response time; s = seconds.

**Table 2 nutrients-14-05264-t002:** Differences between the 3 breakfast categories.

	Had a Full Breakfast	Did Not Eat or Drink	Skipping Breakfast	*p* Value
	*n* = 135	*n* = 62	*n* = 24	
BMI	20.1 ± 4.9	20.1 ± 4.0	21.0 ± 5.2	0.654
Sleep duration, hours	09:28 ± 0:54	09:00 ± 1:06	09:10 ± 0:49	0.012
Bedtime	20:46 ± 0:49	20:50 ± 2:47	21:12 ± 0:46	0.007
Wake up-time	06:14 ± 0:29	06:14 ± 0:35	06:22 ± 0:24	0.501
Non-school PA frequency	4 times/week	3 times/week	2 times/week	0.014
Sedentary within 24 h, %	45.0	44.7	46.7	0.671
Step counts within 24 h	15,884	15,978	14,419	0.037
**Visual Attention Task**				
First RT (s)	4.4 ± 1.9	4.1 ± 0.9	3.9 ± 1.0	0.364
Half-time of RT (s)	22.0 ± 4.6	21.2 ± 4.3	21.3 ± 4.4	0.549
Total RT (s)	40.6 ± 8.3	38.9 ± 8.7	39.0 ± 7.2	0.450

Kruskal–Wallis test; data are presented as median or average values ± standard deviation; *n* = number; PA = physical activity; RT = response time; s = seconds; *p* ≤ 0.05.

## Data Availability

Data are contained within the article.

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
