# Peer review of "Unfavorable Behaviors in Children Run in Packs! Dietary and Non-Dietary Modulators of Attentional Capacity"

_nutrients, 2022, doi:10.3390/nu14245264_

Round 1
Reviewer 1 Report
Thank you very much for the opportunity to read the text.
The article is very well written. It contains all the elements that allow it to be described, as scientific.
It is written briefly, concisely and to the point.
The authors touched on the very important topic of eating the most important meal for teenage children - breakfast, which gives a boost of energy for the whole day.
The illustrations presented in the text add variety to the text, further attracting the reader's attention.
In my opinion, the study is worth publishing.
The only reservation is the small number of literature references only 26 items.
I am interested in whether this is related to the weak interest in the given topic by other researchers?
The purpose is not included in the abstract.
Neither the research questions nor the hypotheses for which answers are sought are marked.
I only found the purpose in the Discussion.
Please correct and complete this.
Author Response
Response to Reviewer 1 Comments
We thank the reviewer for her/his thorough and careful evaluation of our paper and for the constructive critique, comments, and suggestions. We considered all issues addressed and revised our paper accordingly.
Thank you very much for the opportunity to read the text. The article is very well written. It contains all the elements that allow it to be described, as scientific. It is written briefly, concisely and to the point. The authors touched on the very important topic of eating the most important meal for teenage children - breakfast, which gives a boost of energy for the whole day. The illustrations presented in the text add variety to the text, further attracting the reader's attention. In my opinion, the study is worth publishing.
Point 1
The only reservation is the small number of literature references only 26 items. I am interested in whether this is related to the weak interest in the given topic by other researchers?
Thank you for the question! The number of 26 references is a result of the large number of reviews and international reports (12 references are listed below) considered for this brief assessment. Moreover, there are fewer studies of healthy children on the associations between skipping breakfast, other lifestyle characteristics, and cognitive ability. So, we hope that we can make a small contribution with this report.
Reference list from the manuscript: 1, 2, 4, 5, 6, 7, 10, 11, 13, 14, 23, 25.
Point 2
The purpose is not included in the abstract.
We believe that the study purpose/aim stated in the abstract is sufficient for this section given the limited number of 200 characters: Line 11-12: The aim of this analysis was to investigate the mediating effects of breakfast behavior and other lifestyle habits on eye-hand coordination and attention.
Point 3
Neither the research questions nor the hypotheses for which answers are sought are marked. I only found the purpose in the Discussion. Please correct and complete this.
We agree with you that the research questions and hypotheses are missing. The missing description is inserted. Now Line 62-64: The question of the study was whether certain behaviors accumulate in children and correlate with cognition. It is hypothesized that skipping breakfast is related to other unfavorable behaviors in children.
Reviewer 2 Report
In the introduction, the abbreviation PA is used without first defining physical activity. Please correct.
The manuscript needs revisions to correct spelling and grammatical errors. What spelling convention is being used? Line 14 questionnaire spelled incorrectly Line 19 no hyphen in bedtime Line 39 delete "in" and add during the Covid-19 pandemic Line 40 add physical activity and put PA in () Line 44 delete physical activity and replace with PA Line 53 comma after capacity Line 67 analyzed with a Z depending on what spelling convention is being used Line 84 add the after number Line 97 Comma after (Half-time of RT) Line 99 Remove tonight Line 113 add the after were Line 128 space after ' Line 139 replace less with fewer Line 142 sedentary behavior was over... Line 154 correct bedtimes Line 217 replace of with between Correct bedtime throughout the document Line 262 in the first.. Line 281 screen time is 2 words Line 292 correct rarely Line 293 correct well-being
Author Response
Response to Reviewer 2 Comments
We thank the reviewer for the thorough evaluation of our paper and for the suggestions. We have taken into account all the points raised and revised our paper accordingly.
Point 1
In the introduction, the abbreviation PA is used without first defining physical activity. Please correct.
We added the missing description.
Point 2
The manuscript needs revisions to correct spelling and grammatical errors. What spelling convention is being used? We have been using American English. We have checked the document again for spelling errors.
Line 14 questionnaire spelled incorrectly. We have added "n" accordingly.
Line 19 no hyphen in bedtime. It has been corrected in line 19, as well as throughout the manuscript.
Line 39 delete "in" and add during the Covid-19 pandemic. It has been corrected.
Line 40 add physical activity and put PA in (); Line 44 delete physical activity and replace with PA. We have corrected the abbreviation.
Line 53 comma after capacity. It has been inserted.
Line 67 analyzed with a Z depending on what spelling convention is being used. "Z" in regards to American English has been corrected.
Line 84 add the after number; Line 97 Comma after (Half-time of RT); Line 99 Remove tonight; Line 113 add the after were; Line 128 space after '; Line 139 replace less with fewer; Line 142 sedentary behavior was over...; Line 154 correct bedtimes; Line 217 replace of with between. All errors have been fixed.
Correct bedtime throughout the document. It has been corrected throughout the manuscript.
Line 262 in the first..; Line 281 screen time is 2 words; Line 292 correct rarely; Line 293 correct well-being. All corrections have been made.